# Circulating Plasma Exosomal Proteins of Either SHIV-Infected Rhesus Macaque or HIV-Infected Patient Indicates a Link to Neuropathogenesis

**DOI:** 10.3390/v15030794

**Published:** 2023-03-21

**Authors:** Partha K. Chandra, Stephen E. Braun, Sudipa Maity, Jorge A. Castorena-Gonzalez, Hogyoung Kim, Jeffrey G. Shaffer, Sinisa Cikic, Ibolya Rutkai, Jia Fan, Jessie J. Guidry, David K. Worthylake, Chenzhong Li, Asim B. Abdel-Mageed, David W. Busija

**Affiliations:** 1Department of Pharmacology, Tulane University School of Medicine, New Orleans, LA 70112, USA; 2Division of Immunology, Tulane National Primate Research Center, Covington, LA 70433, USA; 3Department of Biochemistry and Molecular Biology, Tulane University School of Medicine, New Orleans, LA 70112, USA; 4Department of Urology, Tulane University School of Medicine, New Orleans, LA 70112, USA; 5Department of Biostatistics and Data Science, Tulane University, New Orleans, LA 70112, USA; 6Proteomics Core Facility, Louisiana State University, New Orleans, LA 70112, USA

**Keywords:** HIV-1, SHIV, circulating plasma exosomes, neuropathogenesis, rhesus macaque, proteomic analysis

## Abstract

Despite the suppression of human immunodeficiency virus (HIV) replication by combined antiretroviral therapy (cART), 50–60% of HIV-infected patients suffer from HIV-associated neurocognitive disorders (HAND). Studies are uncovering the role of extracellular vesicles (EVs), especially exosomes, in the central nervous system (CNS) due to HIV infection. We investigated links among circulating plasma exosomal (crExo) proteins and neuropathogenesis in simian/human immunodeficiency virus (SHIV)-infected rhesus macaques (RM) and HIV-infected and cART treated patients (Patient-Exo). Isolated EVs from SHIV-infected (SHIV-Exo) and uninfected (CTL-Exo) RM were predominantly exosomes (particle size < 150 nm). Proteomic analysis quantified 5654 proteins, of which 236 proteins (~4%) were significantly, differentially expressed (DE) between SHIV-/CTL-Exo. Interestingly, different CNS cell specific markers were abundantly expressed in crExo. Proteins involved in latent viral reactivation, neuroinflammation, neuropathology-associated interactive as well as signaling molecules were expressed at significantly higher levels in SHIV-Exo than CTL-Exo. However, proteins involved in mitochondrial biogenesis, ATP production, autophagy, endocytosis, exocytosis, and cytoskeleton organization were significantly less expressed in SHIV-Exo than CTL-Exo. Interestingly, proteins involved in oxidative stress, mitochondrial biogenesis, ATP production, and autophagy were significantly downregulated in primary human brain microvascular endothelial cells exposed with HIV+/cART+ Patient-Exo. We showed that Patient-Exo significantly increased blood–brain barrier permeability, possibly due to loss of platelet endothelial cell adhesion molecule-1 protein and actin cytoskeleton structure. Our novel findings suggest that circulating exosomal proteins expressed CNS cell markers—possibly associated with viral reactivation and neuropathogenesis—that may elucidate the etiology of HAND.

## 1. Introduction

Globally, there are ~37.5 million people infected with human immunodeficiency virus (HIV), with 1.5 million new infections in 2020 alone [1]. Although the effective combined antiretroviral therapy (cART) has significantly reduced HIV-associated deaths, ~50–60% of patients may develop HIV-associated neurocognitive disorder (HAND) [2]. The pathological causes of HAND are still under investigation. 

Recently, extracellular vesicles (EVs) have had increasing prominence in physiology and pathology research due to their ability to package and deliver payloads (lipids, proteins, or nucleic acids) and molecular signals, enabling cell-to-cell communication and tissue homeostasis [3,4], or to stimulate pathogenesis in numerous diseases [5,6]. Recently, attention has been aimed toward discovering the role of EVs in HIV infection in the body and central nervous system (CNS). Once within the CNS, HIV stimulates neuronal damage using a variety of mechanisms via “direct” or “indirect” neurotoxicity [7]. Several studies have reported that viral proteins are the “direct” cause of CNS dysfunction [8,9,10]. We and others have reported that EVs, or exosomes (well-characterized EVs with a size < 150 nm), released by HIV-1 infected cells containing HIV-1 Tat [11], envelope protein (env) [12], gp120 [13,14], HIV-1 Gag [14], the Nef protein [15], and trans-activation response elements [16,17] may induce neuropathogenesis. Conversely, exosomes have been described as an “indirect” mechanism(s) to accelerate the progression of neurodegenerative disorders such as Alzheimer’s disease (AD), Parkinson’s disease (PD), and/or other neurodegenerative diseases through the supply of proteins or molecules associated with the disease pathology [18,19,20,21,22,23]. Recently, we reported that exosomes released by HIV-infected T-cells or monocytes disrupted mitochondrial dynamics and endothelial nitric oxide synthase (e-NOS) function in primary human brain microvascular endothelial cells (HBMVECs) [11]. Using proteomic analysis, András et al. [24] confirmed that HIV and amyloid beta (Aβ) interact to change the EV composition of HBMVEC, with an increase in proteins engaged in exocytosis, vesicle formation, and immune activation. Another proteomic study of EVs isolated from the cerebrospinal fluid (CSF) of HIV-infected patients with clinical manifestation of HAND reported that markers for microglial activation, inflammation, and stress responses were increased in isolated EVs [25,26]. 

To establish neurocognitive dysfunction using magnetic resonance imaging or scan-based position emission topography methods is costly, and these facilities are limited in developing countries. Diagnosis from CSF for this purpose is an invasive and tedious method. There is a need for a noninvasive technique to address the development and progression of neuropathogenesis, especially in AIDS patients. Recent findings showed that exosomes originating from the CNS can cross the blood–brain barrier (BBB) and carry the pathologic proteins into the blood [27]. Therefore, serum/plasma-derived exosomes from patients with neurological disorders such as PD and AD are being examined for promising biomarkers of neuropathogenesis and clinical progression [21,22,23], and recent studies have characterized plasma EVs/exosome abundance and payload in HIV-infected patients [28,29,30]. We used a liquid chromatography/mass spectrometry (LC-MS/MS)-based proteomic approach to examine circulating plasma exosomal protein links to neuropathogenesis in simian/human immunodeficiency virus (SHIV)-infected rhesus macaques (RM). Plasma exosomes isolated from HIV-infected and cART treated patients have validated key proteomic results of circulating exosomes of RM. 

## 2. Materials and Methods

### 2.1. SHIV-Infected and cART-Treated Rhesus Macaque Plasma

The rhesus macaques (RM) used in this study were housed at the Tulane National Primate Research Center (TNPRC). In addition to complying with the Animal Welfare Act enforced by the U.S. Department of Agriculture (USDA) and Office of Laboratory Animal Welfare (OLAW) within the National Institutes of Health, the TNPRC is accredited by the Association for Assessment and Accreditation of Laboratory Animal Care International, which exceeds federal statutes and policies. Animal care followed all institutional guidelines and recommendations. 

Plasma samples from five SHIV-infected and five uninfected control RM were included in this study. For proteome analysis, we included plasma samples from three randomly selected SHIV-infected (N = 3) and three uninfected animals (N = 3). The average ages of the SHIV-infected and uninfected animals were 11.8 years (14.49 y, 9.51 y, and 11.45 y) and 8.7 years (6.69 y, 7.72 y, and 11.83 y), respectively. Animals were challenged intravenously with the pathogenic chimeric SHIV-D (50 ng of SIV p27 Gag), which is an R5/X4 tropic virus, like many strains of HIV-1. After plasma viral load (PVL) reached viral set point (~6 weeks post-infection), animals were treated daily s.q. for ~12 weeks with a triple-drug ART regimen (including two nucleotide analogs, Tenofovir (TDF) and Emtricitabine (FTC), to inhibit reverse transcriptase, and dolutegravir (DTG) to inhibit integrase) to bring PVL to undetectable levels, analogous to many clinical drug regimens. Animals were followed up for 86 weeks, and plasma were collection at the time of necropsy.

### 2.2. HIV-Infected and cART-Treated Patient Plasma Samples

Normal human plasma was purchased from Sigma-Aldrich, St. Louis, MO, USA (Cat # H4522). HIV-infected and cART-treated (Prezcobix 800 mg-150 mg, Tivicay 50 mg, Etravirin 200 mg) patient plasma was purchased from Innovative Research (Peary Court, Novil, MI, USA). 

### 2.3. RNA Isolation and Viral Load Quantification by qRT-PCR

SHIV viral load was quantified at the Tulane National Primate Research Center Pathogen Detection and Quantification Core, as previously published [31]. Briefly, RNA was extracted from 500 µL of plasma or exosomes suspension with the High Pure Viral RNA kit (cat. #11858882001; Roche, Indianapolis, IN, USA) according to the manufacture’s protocol. Reverse transcription was performed with specific primers and quantitative Real-Time PCR was performed in duplicate with primers and probes specific for the gag gene, and armored RNA for hepatitis C virus was used as internal positive control.

### 2.4. Isolation of Exosomes from Rhesus Macaque (RM) and Human Plasma 

The EVs were isolated from 4 mL of SHIV+/− RM and 1 mL of HIV+/− human plasma by the exoEasy Maxi Kit (QIAGEN, 40724 Hilden, Germany, Cat # 76064) according to the manufacturer’s protocol. Initially, plasma samples were differentially centrifuged at 500× *g* for 5 min, 2000× *g* for 10 min and 10,000× *g* for 30 min to remove cells, cellular debris, and large microvesicles, respectively. Plasma was passed through a syringe filter (Millipore Millex-AA; Cat. # SLAAR33SS) to exclude particles larger than 0.8 µm. Following manufacturer instruction, we added 1 volume of buffer XBP to 1 volume samples, mixed by gently inverting the tubes 5 times, and samples were kept for 15 min at room temperature. The sample/XBP mix was added onto the exoEasy spin column, centrifuged at 500× *g* for 1 min to remove the residual liquid from the membrane, and then centrifuged at 3000× *g* for 1 min. Then, 10 mL of buffer XWP was added and centrifuged at 3000× *g* for 5 min. Finally, EVs were eluted at 600 and 400 µL of XE buffer for RM and human samples, respectively. All the isolation steps were performed at room temperature. The isolated EVs protein concentration was measured using BCA protein assay kit (Pierce, Therma Scientific, Rockford, IL, USA), and were aliquoted with 50 µL of volume and stored at −80 °C until used. 

### 2.5. Characterization of EVs by the ZetaView Particle Metrix System

Size and concentration of the isolated extracellular vesicles were measured using a ZetaView PMX-430-Z QUATT laser system 405/488/520/640 with fixed cell assembly for size, concentration, and zeta potential (Particle Metrix, Mebane, NC, USA) and the corresponding software ZetaView v8.05.16 SP3. Briefly, the system was calibrated and aligned with diluted (1:250,000) 100 nm polystyrene standard polymer particles in aqueous suspension (Applied Microspheres, Leusden, The Netherlands) before the experiment. Samples were kept at room temperature for 30–45 min to accustom before measurement. Samples were diluted to appropriate concentration (1:1000) in deionized-distilled water (ddH_2_O) to reach particle numbers ideal for ZetaView Particle Metrix system. All samples were analyzed under the same conditions (room temperature nearly 20 °C to 25 °C, pH 7.0, sensitivity 80, shutter speed 100, and each replicate included eleven positions). 

### 2.6. Quantitative Discovery-Based Proteomics Using Tandem Mass Tags (TMT) and Liquid Chromatography Mass Spectrometry (LC-MS) 

Samples were prepared for discovery-based quantitative proteomic analysis as previously described [32,33,34]. Briefly, the tandem mass tag (TMT) approach uses isobaric tags to differentiate multiplexed protein extracts using LC-MS. Complexity is reduced by the incorporation of an off-line fractionation step: a basic pH reverse-phase liquid chromatography. The fractionated labeled peptide mixtures were run on a Dionex U3000 nano-flow system coupled to a Thermo Fisher Fusion Orbitrap mass spectrometer. Each fraction was subjected to a 95 min chromatographic method employing a gradient from 2–25% ACN in 0.1% formic acid (FA) (ACN/FA) over the course of 65 min, a gradient of 50% ACN/FA for an additional 10 min, a step to 90% ACN/FA for 5 min, and a 15 min re-equilibration to 2% ACN/FA. Chromatography was conducted in a “trap-and-load” format using an EASY-Spray source (Thermo, Rockford, IL, USA); trap column C18 PepMap 100, 5 µm, 100 A and the separation column was an EASY-Spray PepMap RSLC C18 2 µm, 100A, 75 µm × 25 cm (Thermo Fisher Dionex, Sunnyvale, CA, USA). The entire run was at a flow rate of 0.3 µL/min.

TMT data acquisition utilized an MS3 approach for data collection, where survey scans (MS1) were performed in the Orbitrap utilizing a resolution of 120,000. Data-dependent MS2 scans were performed in the linear ion trap using a collision-induced dissociation (CID) of 25%, and reporter ions were fragmented using a high-energy collision dissociation (HCD) of 55%, detected in the Orbitrap using a resolution of 30,000 (MS3). The LC-MS acquisition data were searched using the SEQUEST HT node of Proteome Discoverer 2.4 (PD) (Thermo). The protein FASTA database was *Rhesus macaque*, SwissProt tax ID = 9544, version 2021-05-24 containing 44,389 sequences. Static modifications included TMT reagents on lysine and N-terminus (+229.163), carbamidomethyl on cysteines (+57.021), dynamic serine, threonine, and tyrosine phosphorylation (+79.966 Da), as well as dynamic modification of methionine oxidation (+15.9949). Parent ion tolerance was 10 ppm, fragment mass tolerance was 0.6 Da for MS2 scans, and the maximum number of missed cleavages was set to 2. 

### 2.7. Cell Culture

Primary HBMVECs were purchased from Cell Systems, Kirkland, WA, USA and were cultured as described previously [11]. Briefly, cells were cultured in complete classic medium with 10% fetal bovine serum and culture boost (4Z0-500) and Bac-Off (4Z0-644). For cell propagation, passage reagent group (4Z0-800) was used according to the recommendations. Briefly, HBMVECs were washed with passage reagent group-1 (PRG-1), dissociated with PRG-2, and the enzymatic reaction stopped with ice-cold PRG-3. Cells were centrifuged at 200× *g* for 7 min at 4 °C. Culture boosts were added to the pellet, and the cells were resuspended in the complete classic medium. Cells were seeded 1:3 or 1:5 in T75 flasks coated with attachment factor (4Z0-210) and incubated at 37 °C with 5% CO_2_ in 95% relative humidity. Cells were fed with fresh media every 48 h and were used up to passage 9. 

### 2.8. Antibodies and Chemicals

Antibodies were purchased from the following suppliers: against Catalase (#14097), phospho-DRP1 (#3455), GAPDH (#2118), CORO1A (#92904), and LC3B (#2775) from Cell Signaling Technology, Danvers MA, USA; against CD81 (#sc-23962), CD63 (#sc-5275), and Flotillin-1 (#sc-25506) from Santa Cruz Biotechnology, Dallas, TX, USA; against β-actin (#A5441) from Sigma-Aldrich, St. Louis, MO, USA; against GFAP (#610565) from BD Transduction Laboratory, San Jose, CA, USA; against von Willebrand factor (VWF) (#ab6994), PECAM-1 (#ab28364) from Abcam, Cambridge, MA, USA; anti OMG (#12701-1-AP) and GABRA1 (#12410-1-AP) from Proteintech Group, Inc., Rosemont, IL, USA; and mitochondrial complex III subunit core 1 from Invitrogen Corporation, Fredrick, MD, USA. 

### 2.9. Western Blotting

We followed our previously described standard laboratory protocol to prepare the lysates from cell/exosome and to perform the immunoblots [11,32,33]. In brief, phosphatase and protease inhibitors containing ice-cold NP40 lysis buffer (Invitrogen, Frederick, MD, USA) was used to lyse either cells or exosomes. The clarified lysate protein concentration was measured by Pierce BCA protein assay (Thermo Scientific). The proteins were separated using a 4–20% SDS-PAGE gradient gel and transferred onto a PVDF membrane. A 1X blocking buffer (Abcam, Cambridge, MA, USA) was used to block the non-specific binding sites, and to dilute the primary antibodies. The membranes were washed with Tris-buffered saline (Bio-Rad, Hercules, CA, USA) containing 0.1% Tween-20 (Sigma-Aldrich, St. Louis, MO, USA). Membranes were incubated overnight with primary antibodies at 4 °C, then membranes were washed and incubated again with respective secondary antibodies, either goat anti-rabbit IgG at 1:2500 dilution (#7074S, Cell Signaling Technology) or goat anti-mouse IgG at 1:5000 dilution (#7076P2, Cell Signaling Technology) at room temperature for 1 h. Chemiluminescence (LumiGLO, Gaithersburg, MD, USA) and autoradiography were used to visualize the final reaction. In some cases, immunoblot signals were captured using the ImageQuant Las 300 (GE Healthcare, Piscataway, NJ, USA) system. Immune band densitometry was performed using ImageJ Software (NIH, Bethesda, MD, USA, http://imagej.nih.gov/ij/ accessed on 4 November 2021).

### 2.10. Transwell Primary Brain Endothelial Cell Permeability Assay

Conventional in vitro BBB modeling was conducted by culturing primary HBMVECs grown in 2D as a flat monolayer by following the published protocols [35]. Permeability across the monolayer of HBMVECs was measured by using transwell units (24-well format, 0.4 µm pore size with high pore density PET track-etched membrane, cat # 353495), which were made to fit in a 12-well plate, purchased from Corning Incorporated, Corning, NY, USA. To create the BBB model, 1 × 10^5^ cells were seeded onto the transwell inserts and cells were allowed to grow for 2 days in complete classic medium. After endothelial cells become confluent on the transwell, cells were treated in fresh culture medium with 10 µg/mL of either Patient-Exo or Control-Exo for 24 h. The cell monolayer’s ability to limit the penetration of a high molecular weight compound was measured by adding 10 µg/mL fluorescein isothiocyanate-labeled dextran (FITC-dextran; molecular weight: 150 kDa, Sigma-Aldrich, St. Louis, MO, USA) to the culture transwell inserts. In the absence of primary HBMVEC, FITC-dextran added to the culture inserts served as controls. After incubation for 10, 30, 60, and 120 min, 20 µL of medium was collected from the lower compartment, and the fluorescence was measured with a spectrophotometer (BioTek Instruments, Winooski, VT, USA) set to 485/20 nm excitation and 528/20 nm emission. 

### 2.11. Immunofluorescence, Confocal Microscopy, and Image Analysis 

After experimentation, cultured HBMVECs were fixed with 4% PFA for 15 min at room temperature and washed three times with PBS (5 min per wash). Prior to staining, fixed cells were permeabilized with PBST (PBS with 0.1% Triton X-100) for 20 min. For staining using antibodies, cells were blocked for non-specific binding using PBS containing 5% donkey serum (Sigma-Aldrich Cat. No.: D9663) for 1 h at room temperature, followed by overnight incubation at 4 °C with a rabbit PECAM-1 (CD31) polyclonal antibody (BiCell Scientific Cat. No.: 01004) at a 1:200 dilution in PBS with 5% donkey serum. The following morning, cells were rinsed once with PBS, and then washed for 2 h at room temperature on a rocker, replacing the PBS 2–3 times during this period. Cells were then incubated, protected from the light with a donkey anti-rabbit IgG(H+L) highly cross-adsorbed secondary antibody conjugated with Alexa Fluor 488 (Invitrogen Cat. No.: A-21206) at a 1:500 dilution for 2 h at room temperature on a rocker. Finally, cells were rinsed with PBS once to remove excess unbound secondary antibody, then washed for 2 h at room temperature on a rocker, with the PBS replaced 2–3 times during this period, and then mounted using a ProLong Glass antifade mountant with NucBlue stain (Invitrogen Cat. No.: P36981). 

For F-actin filament staining, after fixation and permeabilization described above, cells were incubated with Alexa Fluor 488 Phalloidin (Invitrogen Cat. No.: A12379, 300 U, stock concentration 400×) at a 1:400 dilution in PBS containing 5% donkey serum for 30 min at room temperature on a rocker. Then, the cells were rinsed once with PBS, and then washed for 20 additional min with the same solution on a rocker and protected from light. After staining, cells were mounted as described above. Antifade mountant was allowed to cure overnight at room temperature, protected from light. Fluorescence images were collected using a 40× water immersion objective on an Andor Dragonfly 202 (+Leica DMI8) high speed confocal imaging platform equipped with solid state 405 nm and 488 nm smart diode lasers and a Zyla PLUS 4.2 Megapixel sCMOS camera. Using ImageJ Fiji, the raw integrated density for each immunofluorescence image was calculated. Then, using the DAPI staining channel, the number of nuclei were determined via particle analysis. Finally, the raw integrated density calculations were normalized by the number of cells/nuclei. Data are presented as the mean ± SEM of the normalized integrated fluorescence density for each group with the control experiments mean as the reference, i.e., mean of control group being 100 percent. Statistically significant differences were evaluated using either parametric *t* tests or a one-way ANOVA test with Tukey post hoc, * *p* < 0.05. 

### 2.12. Statistical Analysis

Only high scoring peptides were included for proteomic analysis, using a false discovery rate (FDR) of <1%. Only one unique high-scoring peptide was essential for inclusion as an identified protein in our results. Proteome Discoverer was also used to determine quantitative differences between biological groups. Quantitative data were collected using a *t*-test analysis on grouped biological replicates and we performed pair-wise comparisons for fold-change. The normalized abundance quantity of a biological replicate was calculated by averaging four experimental replicates. The data are presented as mean ± standard deviation (SD). For normal distribution, the data sets were assessed by the Shapiro–Wilk or Kolmogorov–Smirnov test, followed by unpaired t test with Welch correction for normally distributed data. When the data did not pass the normality test, a non-parametric Mann–Whitney test was performed, as indicated in the figure legends. The statistical analysis was performed using GraphPad Prism version 9.0.0 for Windows, and *p* < 0.05 was considered statistically significant. Hierarchical clustering analysis (HCA) was performed in R Studio (version 1.4.1103) using “Manhattan” clustering and “complete” linkage method.

## 3. Results

### 3.1. Characterization of Circulating Plasma Exosomal Proteome of SHIV-Infected and Uninfected RM

Initially, we measured the viral load in plasma and in isolated exosomes (crExo) by qRT-PCR. The plasma qRT-PCR results indicated that two SHIV+ samples had low viral copy numbers (3.1 and 3.4 equivalent (Eq.) LOG copies) and one SHIV+ sample was less than 1.9 Eq. LOG copies (the detection limit). All uninfected (N = 3) plasma samples were negative. In the exosomal fraction, the viral load was less than 1.9 Eq. LOG copies in the SHIV+ samples (N = 3), and negative in uninfected samples (N = 3) (data not shown). Although we isolated exosomes from the commercially available exosome isolation kit, we further confirmed the size (nm) and concentration (particles/mL) by the ZetaView analyzer of SHIV-/CTL-Exo (N = 5/group). Interestingly, both the size and concentration of SHIV-Exo were significantly lower than CTL-Exo (Figure 1a,b). By proteomic analysis, a total of 5654 proteins were quantified, of which 236 proteins (~4%) were significantly, differentially expressed (DE) between SHIV-/CTL-Exo (N = 3/group) (Figure 1c). Two or more unique peptides were detected in 85% (4777/5654) of quantified proteins, and in 89% (211/236) of significant DE proteins, indicating the depth of analysis (Figure 1d,e). The hierarchical cluster analysis (HCA) of significant DE proteins were presented in Figure 1f. Moreover, the HCA of top 50 significant up-/down-regulated proteins in SHIV-Exo were presented in Figure 2a,b. We used PCA to decrease the data dimensions for simpler interpretation. According to PCA analysis on all protein expression data, we observed that the CTL-Exo (in red) are clustered separately from the SHIV-Exo (in green). The percentage of variance indicates how much variance was explained by principal component 1 (PC1) and principal component 2 (PC2) and it was 25.2% and 21.6%, respectively. Thus, our preliminary proteome comparisons reveal that there are substantial differences between CTL-Exo and SHIV-Exo (Figure 2c).

### 3.2. Hallmark Exosomal Proteins Were Quantified by Proteomic Analysis in crExo of SHIV-Infected and Uninfected RM

Like a typical cell membrane, the exosomal membrane consists of lipids and proteins. Tetraspanin proteins CD9, CD63, CD81, CD82, CD151, Tetraspanin-5 (TSPAN5), TSPAN9, and TSPAN14 were abundantly expressed in both SHIV-Exo and CTL-Exo (Figure 3a). Several cytoskeletal proteins were quantified in both SHIV-Exo and CTL-Exo. The relative abundance of actin alpha 1, skeletal muscle (ACTA1), tubulin alpha chain (TUBA1B), tubulin beta chain (TUBB), myosin light chain 2 (MYL2), myosin heavy chain 11 (MYH11), vimentin (VIM), and radixin (RDX) were low in both SHIV-Exo and CTL-Exo. However, the relative abundance of fibronectin (FN1), ezrin (EZR), keratin 1 (KRT1), and keratin 2 (KRT2) were high in both SHIV-Exo and CTL-Exo. Interestingly, the abundance of RDX was significantly higher in CTL-Exo than SHIV-Exo (Figure 3b). Cytosolic enzymes such as glucose-6-phosphate isomerase (GPI), glyceraldehyde-3-phosphate dehydrogenase (GAPDH), fructose-bisphosphate aldolase A (ALDOA), fructose-bisphosphate aldolase C (ALDOC), 2-phospho-D-glycerate hydro-lyase (gamma enolase; ENO1), and pyruvate kinase PKLR (PKLR) were expressed equally in both SHIV-Exo and CTL-Exo (Figure 3c). The exosomes composition reflects to some extent the composition of multivesicular bodies (MVBs). In fact, proteins associated with MVBs, such as tumor susceptibility 101 (TSG101), clathrin light chain (CLTA), clathrin heavy chain (CLTC), syntaxin-7 (CTX7), and syntaxin-8 (CTX8), were also expressed equally in both SHIV-Exo and CTL-Exo (Figure 3d). Exosomes have been shown to be enriched in cholesterol, sphingomyelin, and glycosphingolipids that play important roles in signaling and sorting. In this study, NPC intracellular cholesterol transporter 1 (NPC1), stomatin-like protein 2 (STOML2), stomatin-like protein 3 (STOML3), flotillin-1 (FLOT1), flotillin-2 (FLOT2), sphingosine kinase 2 (SPHK2), sphingosine 1-phosphate receptor 1 (S1PR1), and sphingosine-1-phosphate lyase 1 (SGPL1) were abundantly expressed in both SHIV-Exo and CTL-Exo (Figure 3e). The CD63, CD81, GAPDH, and FLOT1 expression was further validated by Western blotting (Figure 3f). 

### 3.3. Different CNS Cell Markers Were Abundantly Detected in SHIV-/CTL-Exo

Interestingly, we observed that brain microvascular endothelial cell specific von Willebrand factor (VWF) and platelet and endothelial cell adhesion molecule-1 (PECAM-1), as well as microglial cell markers beta-hexosaminidase (HEXB), spalt-like transcription factor 1 (SALL1), receptor protein-tyrosine kinase (MERTK), and coronin 1A (CORO1A), were differentially expressed in SHIV-/CTL-Exo. In addition, thought provoking, astrocyte cell markers such as glial fibrillary acidic protein (GFAP), protein S100 (S100B), aquaporin-4 isoform a (AQP4), and glutamine synthetase (GLUL) were abundantly less expressed both in SHIV-/CTL-Exo. Two oligodendrocyte markers: myelin basic protein (MBP) and myelin-oligodendrocyte glycoprotein (MOG) were also less expressed in SHIV-/CTL-Exo (but not oligodendrocyte-myelin glycoprotein (OMG)). Moreover, several neuron-specific proteins, including neurofilament heavy (NEFH), neurofilament light polypeptide (NEFL), 160 kDa neurofilament protein (NEFM), synaptophysin (SYP), T-box brain protein 1 (TBR1), stathmin 1 (STMN1), alpha-internexin (INA), axonal membrane protein GAP-43 (GAP43), GABA(A) receptor subunit alpha-1 (GABRA1), and CLPTM1 regulator of GABA type A receptor forward trafficking (CLPTM1), were also quantified in both SHIV-/CTL-Exo (Figure 4a). The expression of several CNS cell markers, specifically VWF, PECM-1, CORO1A, GFAP, OMG, and GABRA1 in SHIV-/CTL-Exo, was further validated by Western blotting (Figure 4b). 

### 3.4. Proteins Involved in Viral Reactivation, Inflammation, and Neuropathology-Associated Interactive/Signaling Proteins Were Significantly Higher in SHIV-Exo Than CTL-Exo

Pan et al. [36] reported that heat shock factor 1 (HSF1) positively regulates transcription of latent HIV infection. We observed a significantly greater increase of HSF1 in SHIV-Exo than CTL-Exo, possibly indicating the positive regulation of latent infection in SHIV-infected RM. Plasma proteins involved in inflammation and complement dysregulation support diagnosis and outcome predictions of mild cognitive impairment and AD. Our proteomic study showed that the abundance of complement factor H (CFH), complement component 8 subunit (C8B), and thioredoxin domain-containing protein (TXN2) were significantly higher in SHIV-Exo than CTL-Exo. Platelet glycoprotein 1b subunit alpha (GP1BA), a thromboinflammatory axis in cardiovascular pathologies, was also significantly higher in SHIV-Exo (Figure 5a). In SHIV-Exo, neuropathology-associated interactive proteins such as amyloid beta precursor protein binding family B number 1 (APBB1), chromogranin A (CHGA), chromogranin B (CHGB), and peptidylprolyl isomerase (FKBP2) were significantly higher in SHIV-Exo than CTL-Exo (Figure 5b). Similarly, neuropathology-associated signaling proteins such as cyclin-H (CCNH), CUGBP Elav-like family member 3 (CELF3), RPTOR independent companion of MTOR complex 2 (RICTOR), CD74, and LIM domain kinase 1 (LIMK1) were also significantly higher in SHIV-Exo than CTL-Exo (Figure 5c). Age-adjusted *p*-value was also significant (*p* < 0.05) for HSF1, CFH, TXN2, APBB1, CCNH, RICTOR, and LIMK1 (Appendix A). 

### 3.5. Proteins Involved in Mitochondrial Biogenesis and ATP Production Were Reduced in SHIV-Exo Than CTL-Exo

Mitochondrial (Mt) antiviral signaling proteins (MAVS) as well as Mt fission-related Mt fission 1 (FIS1), Mt fission factor (MFF), and Mt fission regulator 1 (MTFR1L) proteins were significantly less expressed in SHIV-Exo (Figure 6a). Proteins involved in ATP production such as NADH-ubiquinone oxidoreductase chain 5 (ND5), Mt complex I subunit B13 (NDUFA5), cytochrome c oxidase subunit 4 isoform 2 (COX4l2), cytochrome c oxidase assembly protein COX20 (COX20), ATP synthase subunit g (ATP5L), ATP synthase subunit alpha (ATP5F1A), ATP synthase subunit d (ATP5PD), mitochondrial ribosome recycling factor (MRRF), 2-oxoglutarate dehydrogenase-like, mitochondrial isoform a (OGDHL), 2-oxoglutarate dehydrogenase-like, and mitochondrial isoform a (COQ6) were also significantly less expressed in SHIV-Exo than in CLT-Exo (Figure 6b,c). Age-adjusted *p*-value was also significant (*p* < 0.05) for MAVS, FIS1, MTFR1L, NDYFA5, COX4l2, COX20, ATP5L, ATP5F1A, MRRF, and COQ6 (Appendix A).

### 3.6. Proteins Involved in Autophagy, Endosomal Recycling, Exocytosis, Sprouting Angiogenesis, Cytoskeleton Organization, and Vesicle Transport Are Downregulated in SHIV-Exo

There is increasing evidence indicating that defective autophagy-lysosome pathways are associated with neurodegenerative diseases. Lysosomal associated membrane protein 2 (LAMP2) has been recognized as a receptor for the selective import and degradation of cytosolic proteins in the lysosome, or chaperone-mediated autophagy [37]. In this study, the expression of LAMP2 was significantly impaired in SHIV-Exo. Moreover, the autophagy-related protein 2 homolog B (ATG2B), and autophagy-related protein 9A (ATG9A) were significantly less expressed in SHIV-Exo. Similarly, CDGSH iron-sulfur domain-containing protein 2 (CISD2), an autophagy regulator, was also significantly less expressed in SHIV-Exo (Figure 7a). The expression of several Ras-related proteins: Rab-3C (RAB3C), -5C (RAB5C), and -13 (RAB13) were significantly less expressed in SHIV-Exo (Figure 7b) as were proteins involved in endosomal recycling: (sorting nexin 4; SNX4), exocytosis (synaptogyrin; SYNGR2), sprouting angiogenesis (Jumonji domain-containing 6, arginine demethylase and lysine hydroxylase; JMJD6), cytoskeleton organization (calponin; CNN2), and neurobehavioral problem (PHD finger protein 21A; PHF21A) (Figure 7c). Age-adjusted *p*-value was also significant (*p* < 0.05) for LAMP2, CISD2, RAB3, SX4, PHF21A, JMJD6, and SYNGR2 (Appendix A).

### 3.7. HIV-Infected and cART-Treated Patient-Exo Decreased the Expression of ROS Scavengers, BBB- and Autophagy-Related Proteins as well as Proteins Involved in Mitochondrial Fusion and Electron Transport Chain in Primary HBMVECs

The expressions of catalase (CAT) and microtubule-associated proteins 1A/1B light chain 3B type-II (LC3B) were downregulated when cells were exposed with Patient-Exo. Usually, the active autophagy is measured by changes in LC3 localization: tracking the level of conversion of LC3-I to LC3-II provides an indicator of autophagic activity that was significantly impaired in cells exposed with Patient-Exo. In addition, the expressions of phosphorylated dynamin-related protein 1 (pDRP1) and mitochondrial complex-III (MC-II) were downregulated due to exposure with Patient-Exo in primary HBMVEC (Figure 8a,b). 

### 3.8. Patient-Exo Increases in BBB Permeability Possibly Due to Loss of PECAM-1 and Actin Cytoskeleton in Primary HBMVECs

The transwell migration assay indicated that the brain endothelial cell permeability was significantly increased when cells were treated with Patient-Exo compared with controls (Figure 9a,b). Immunofluorescence assay showed that the expression of PECAM-1 was significantly impaired in cells exposed with Patient-Exo (Figure 9c,d). 

Several studies have demonstrated that BBB permeability changes as a result of actin reorganization [38,39,40]. In our proteomic study, several actin-related proteins were abundantly less expressed in SHIV-Exo than in CTL-Exo. Interestingly, the expression actin-depolymerizing factor (GSN) was significantly higher in SHIV-Exo (Figure 10a). Our immunofluorescence study showed that HBMVEC exposed with Patient-Exo lost the typical architecture of filamentous actin (F-actin) and accumulated it within the cells (Figure 10b). 

## 4. Discussion

Extracellular vesicles/exosomes have been involved in regulating the progression of various neurodegenerative diseases by supplying pathogenic proteins or molecules associated with such diseases [18,19,20,21,22,23]. Recent studies have demonstrated the ability to isolate brain-generated exosomes from blood, and exosomes seem to be minimally invasive biomarkers for various neurodegenerative diseases [41]. Moreover, findings have established the possible role of exosomes as diagnostic biomarkers of HAND [25,26,42]. Using proteomics, Guha et al. [25] reported that some proteins in CNS cell-specific exosomes were higher in HIV patients with HAND compared with those without HAND. In this study, we have characterized the circulating plasma exosomal (crExo) proteins in SHIV-infected and uninfected RM and their possible link with neuropathogenesis. We applied discovery-based TMT-tag mass spectrometry analysis to compare the relative crExo protein expression in infected and uninfected RM. In the last decade, there has been an exponential increase in the use of discovery-based proteomic approaches in translational research, due largely to the progress of state-of-the-art mass spectrometry (MS/MS). The measurement accuracy is pronounced, and it is possible to identify and quantify peptides and proteins in a sensitive and high-throughput manner. The quantification of the total number of proteins in exosomes has been different in these studies. Recently, Dr. Okeoma and her research group quantified 625 proteins with higher-stringency (≤1% FDR, and ≥2 unique peptides) from HIV(+)/(−) plasma exosomes [43]. However, in a recent CSF EVs proteomic study, more than 2700 proteins were identified in HIV-infected patients [25]. The chemical labeling methods: either ITRAQ (isobari tags for relative and absolute quantification)- or TMT-tags, which allow high multiplexing are better-suited for multiple samples [44], and this method allows for the identification of even low abundance proteins. This strategy is successfully reflected in our TMT-tag LC-MS/MS analysis and we quantified 5654 proteins (<1% FDR) and two or more unique peptides were detected in 85% of quantified proteins. 

CNS cell-specific proteins were differentially expressed in plasma crExo. Several exosomal hallmark proteins (tetraspanins, enzymes, lipid rafts, cytoskeletal, and endosome-specific proteins) that were abundantly expressed in both CTL-/SHIV-Exo indicated the successful isolation of crExo from plasma. The vesicle size, shape, concentration, and presence of exosomal markers exhibited similarities that were isolated from either human serum or plasma, suggesting that serum and plasma are equally useful for isolation of blood exosomes [45]. Exosomes originating in the CNS can cross the BBB, can be isolated from the blood [46] and are an attractive source to identify ongoing disease progression in the CNS [47,48]. Our study supports this concept. Brain microvascular endothelium-, microglia-, astrocyte-, oligodendrocyte-, and neuron-specific proteins were either quantitatively or qualitatively measured in crExo.

Upregulated proteins in SHIV-Exo may link to HIV-associated neuropathogenesis. HIV infection in the CNS results from transmigration of infected CD4+ T cells and/or monocytes through the BBB. However, these migrated cells in the CNS do not constitute an HIV reservoir since they have a very short half-life. On the other hand, microglial cells, with their long half-life of years [49], are highly susceptible to HIV infection [50,51], support productive infection [52], and serve as a major CNS reservoir of latent provirus. Moreover, inadequate cART supply in CNS contributes to the persistence of infection in microglial cells [53]. Using a macaque model for AIDS and HAND, Clements et al. [54] reported that more than 80% of infected animals develop simian immunodeficiency virus (SIV)-associated neurological symptoms within 90 days. As a host factor, HSF1 significantly contributes to HIV transcription and is essential for HIV latent reactivation [36]. In our study, the abundance of HSF1 was significantly higher in SHIV-Exo than in CTL-Exo, possibly promoting productive infection and deleterious neuroinflammation in CNS. 

HIV both directly and indirectly increases neuroinflammation and the development of clinical symptoms of HAND. Several studies have shown that exosomes transport both viral and host proteins that facilitate neuroinflammation [55,56,57]. Inflammation and complement dysregulation were important components of AD pathogenesis. To identify inflammatory plasma biomarkers in mild cognitive impairment (MCI) and AD patients, it has been reported that complement factor H (CFH), complement component C3, and C5 were significantly higher in MCI (*n* = 199) than control subjects (*n* = 259). Additionally, plasma soluble complement-receptor 1, -C4, and -C5 were significantly high in AD patients (*n* = 262) [58]. Moreover, the “thrombo-inflammatory” nature of VWF-GP1BA axis becomes progressively identified in different cerebrovascular pathologies [59]. Interestingly, in our study, plasma exosomal CFH and C8B were significantly higher in SHIV-Exo than CTL-Exo. We also observed that both VWF and GP1BA were significantly high in SHIV-Exo. Another common feature of neurodegenerative diseases is oxidative stress [60]. Antioxidant TXN2 is an important cellular component against oxidative stress, which we showed was highly expressed in SHIV-Exo, possibly linked to neuropathogenesis documented in previous studies [61,62,63].

One manifestation of AD is the accumulation of amyloid β (Aβ) peptides in the brain originating from the amyloid precursor protein (APP). APBB1 is an adapter protein that forms a transcriptionally active complex with APP and plays a vital role in Aβ binding and positive regulation of the apoptotic process. We found that APBB1 was significantly upregulated in SHIV-Exo. There is a strong correlation between APP, secretory neuropeptides, CHGA, and CHGB [64]. CHGA induces a neurotoxic phenotype in brain microglial cells [65] and may directly modulate synaptic activity [66]. Using postmortem brain of AD patients, Lechner et al. [67] reported that ~30% of Aβ plaques were co-labelled with CHGA (mostly found in pyramidal neurons), and 15% with CHGB (largely located in interneuron). CHGA is concentrated in the Levy bodies in PD in the Pick bodies, and in the swollen neurons in Pick’s disease [68,69] and tends to accumulate in the senile and pre-amyloid plaques in AD [70]. In addition, elevated levels of CHGA have been measured in the CSF of AD patients [71]. Now it is known that EVs/exosomes can transport their cargo bidirectionally from the brain to the systemic circulation. In our study, both CHGA and CHGB were significantly higher in SHIV-Exo than CTL-Exo, indicating they are increased in the infected RM and possibly associated with the progression of HAND.

The CNS is an important HIV reservoir, and CCNH regulate transcription by RNA polymerase II and participate in HIV-1 early elongation complex formation, supporting viral infection and replication in the CNS. Moreover, mTOR activity of RNA-binding protein CELF3 in tau splicing may be a factor promoting neuronal pathology [72]. The signaling protein RICTOR is critical to mTOR function, and Aβ accumulation is associated with an increase in mTOR signaling in postmortem AD brain tissues [73]. Bryan et al. [74] reported that the expression of CD74 is increased in neurofibrillary tangles of AD patients. Aβ42 peptides caused spine degeneration and neuronal hyperexcitability via LIMK1-dependent mechanisms in rat hippocampal neurons [75] and a significant increase of phosphorylated LIMK1-positive neurons was identified in areas affected with AD pathology [76]. We found that the expression of CCNH, CLF3, RICTOR, CD74, and LIMK1 was significantly high in SHIV-Exo, indicating their possible link to neuropathogenesis. 

Downregulated proteins in SHIV-Exo may indicate CNS health imbalance. Mitochondrial function is crucial to organs with a high metabolic rate such as the brain [77]. Several mitochondrial proteins have been found in exosomes. Both Caicedo et al. [78] and Goetzel et al. [79] showed that several mitochondrial proteins including complex-I (subunit 6) and complex-III (subunit 10) were significantly lower in plasma EVs of patients with major depressive disorder than their matched controls. Recently, Chi et al. [80] reported that plasma neuroexosomal NADH ubiquinone oxidoreductase core subunit S3 (NDUFS3) and succinate dehydrogenase complex subunit B (SDHB) levels were significantly lower in AD and in progressive mild cognitive impairment (MCI) than in cognitively normal subjects. Moreover, they observed that plasma neuroexosomal NDUFS3 and SDHB levels were lower in progressive MCI than in stable MCI subjects. Our study supports these findings and we observed that several mitochondrial biogenesis and ETC complex proteins were significantly less expressed both in SHIV-Exo and in HBMVECs exposed with HIV+ Patient-Exo.

Defective autophagy, endosomal recycling, and exocytosis pathways are significantly linked with neuropathogenesis. Guo et al. [81] reported that α-synuclein preformed fibrils decreased autophagy flux due to degradation of LAMP2 in activated microglia. LAMP-2 deficiency leads to hippocampal dysfunction [82]. Axonal autophagosome maturation defect due to failure of ATG9A sorting underlies pathology in adaptor protein-4 deficiency syndrome [83]. Moreover, CISD2 deficiency enhanced AD pathogenesis. Therefore, downregulated expression of LAMP2, ATG9A, and CISD2 in SHIV-Exo and LC3B-II in HBMVECs treated with HIV+ Patient-Exo may be linked to HAND. A subset of RABs is involved in autophagy regulation, including RAB2 [84], RAB3 [85], RAB5 [86], and RAB18 [87]. In our study, RAB3C, RAB5C, and RAB13 were significantly downregulated in SHIV-Exo. RAB-associated dysfunctions are linked with several neuropathological disorders [88,89]. RABs are critical for maintaining neuronal communication, as well as for normal cellular physiology. Therefore, cellular defects of RAB elements severely impact normal brain functions, leading to development of neurodegenerative diseases [89]. In cells, RAB-proteins are inevitable for endocytic and exocytic pathways. Sorting nexins (SNXs) dysfunction has been linked to several neurodegenerative diseases including AD, PD, and Down’s syndrome [90]. SNX4 is expressed in the brain and SNX4 protein levels are decreased by 70% in brains of severe AD cases [91]. Deletion of CNN2 increases macrophage migration and phagocytosis [92]. Disruption of the PHF21A gene triggers syndromic intellectual disability with craniofacial anomalies, and neurobehavioral problems including autism [93]. JMJD6 regulates the VEGF-receptor 1 splicing, thereby controlling angiogenic sprouting [94]. In neuronal cells, SYNGR2 modulates the localization of synaptophysin into synaptic-like MVs and play a role in the formation and/or the maturation of these vesicles. Therefore, downregulated expression of SNX4, CNN2, PHF21A, JMJD6, and SYNGR2 in SHIV-Exo may relate to neuropathogenesis. 

Increased BBB permeability by crExo may be an initiating event for HIV-associated neuropathogenesis. BBB injury is common in patients with HIV-associated dementia and is a frequent feature of HIV encephalitis, although the process is not completely understood [95,96,97]. Together with inflammatory mediators (e.g., cytokines and chemokines), several viral proteins may damage the BBB integrity and facilitate BBB permeability [98]. One study reported that microglia-derived HIV-Nef+ exosomes directly impact BBB integrity and permeability [55], while using HBMVEC, Leon et al. [99] showed that plasma EVs from patients with preeclampsia disrupt BBB integrity. Using a similar approach, we have shown that HIV+ Patient-Exo significantly increased BBB permeability in primary HBMVEC. Interestingly, the expression of ZO-1, ZO-2, and JAM-A was not significantly downregulated (data not shown), but the expression of PECAM-1 was significantly impaired in HBMVECs after exposure to HIV+ Patient-Exo. Using the same BBB model on primary mouse brain microvascular endothelial cells, Wimmer et al. [100] showed that lack of endothelial PECAM-1 increased BBB permeability, although cells maintained intact junctional architecture. In addition, during LPS-induced endotoxemia, blood vessels of PECAM-1-deficient mice exhibited increased BBB permeability [101]. Furthermore, PECAM-1−/− mice displayed impaired BBB integrity and accelerated immune cell infiltration into the CNS [102].

Various studies have also shown that BBB permeability changes due to actin reorganization [38,39,40] and our present findings supports that concept. There is recognition between filamentous actin (F-actin) that makes up the cortical actin ring and stress fibers. During inflammation, stress fibers increase in endothelial cells [103] that increase intracellular tension, reorganize the adhesion complex structure, and generate gaps in the intracellular connections [104], increasing permeability. Therefore, the cortical actin ring is required for adhesion complex structure and BBB integrity [105]. We observed that SHIV-Exo simultaneously expressed a significantly high level of actin depolymerizing factor protein and relatively less expression of several actin cytoskeleton proteins. Moreover, we showed that HBMVECs lose their F-actin cytoskeleton organization once the cells were exposed to HIV+ Patient-Exo. 

Limitations and potentials of their research. There are several limitations in this study. (1) To avoid complexity, we focused only on the host proteins which are possibly linked to HIV-associated neuropathogenesis. However, viral proteins, especially Matrix protein for HIV-1 [106] and other viruses [107], play an important role in viral evasion against antibodies and the ability of HIV-1 Matrix protein p17 to penetrate into the brain possibly influences HAND progression [108]. (2) There are possible differences in the behaviors and properties of SHIV and HIV variants. Therefore, we cannot assume that SHIV-Exo will have the same effect as HIV-infected Patient-Exo on HAND. (3) Small significant differences for some proteins could be the result of variations over a period of time; therefore, we have to focus on a longitudinal study to ensure that cross-sectional results being observed in this study could be replicated at different time points. (4) Since our IRB approval is pending, we only included one of each commercially available patient plasma and healthy donor plasma samples in this study. (5) Several studies reported that monocyte chemoattractant protein-1 (CCL2/MCP-1) in plasma is a marker for HAND [109,110,111]. However, in our study, CCL2/MCP-1 was not quantified in either control or SHIV-infected plasma exosomes. 

## 5. Conclusions

In the brain, EVs/exosomes are engaged in several physiological processes, including BBB permeability, neurogenesis, cell communication, neuronal stress response, synaptic plasticity, and others. Therefore, EVs/exosomes are implicated in the pathology of many neurodegenerative diseases. Many studies have shown the ability of exosomes to isolate brain-generated exosomes from blood, while apparently being minimally invasive biomarkers for various neurodegenerative diseases including HAND. Much is already known about the contribution of CNS cells in HAND. Our research has demonstrated that the role that crExo proteins play in HAND is a promising developing field that may impact the understanding of HIV-associated neuropathogenesis and will help to develop attractive therapeutic options against HAND. 

## Figures and Tables

**Figure 1 viruses-15-00794-f001:**
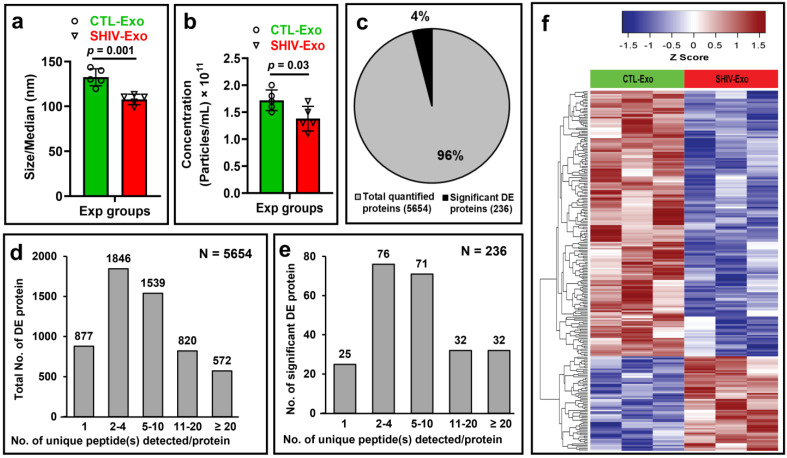
Characterization of circulating plasma exosomal proteome of SHIV-infected and uninfected Rhesus macaque (RM). (**a**,**b**) Exosomes were isolated by exoEasy Maxi Kit from the plasma of SHIV-infected and uninfected RM. The size (nm) and concentration (particles/mL) of the isolated exosomes were characterized by ZetaView Particle Metrix system. (**c**) Total and significantly differentially expressed (DE) proteins in plasma exosomes of SHIV-infected (SHIV-Exo) and control (CTL-Exo) RM quantified by proteomic analysis. (**d**) Comparison of the number of unique peptide(s) detected per quantified proteins. (**e**) Comparison of the number of unique peptide(s) detected in significant DE proteins. (**f**) A hierarchical cluster analysis of significant DE proteins. A heatmap was generated for all the 236 significant DE proteins using “Manhattan” clustering and “complete” linkage method.

**Figure 2 viruses-15-00794-f002:**
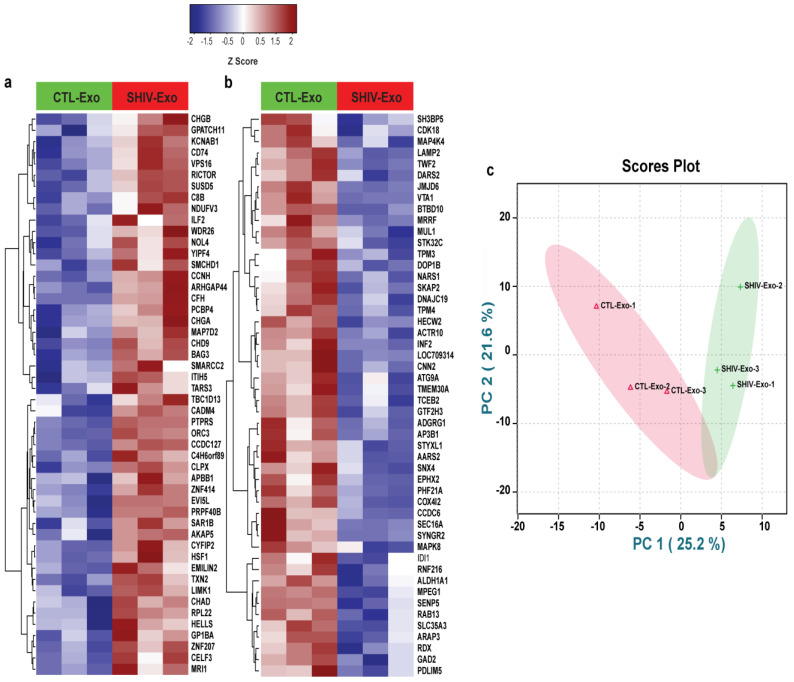
(**a**,**b**) A hierarchical cluster analysis of significant top 50 up-/down-regulated proteins in SHIV-Exo. Proteins were filtered based on *p* < 0.05. Median values were calculated for CTL-Exo and SHIV-Exo to calculate the ratio (SHIV-/CTL-Exo). The ratio was used to generate the list of top 50 up and down regulated proteins. Z scores were calculated for each significant protein, and then heatmaps were generated for all the significant proteins: top 50 upregulated and top 50 downregulated proteins using “Manhattan” clustering and “complete” linkage method. Protein coding gene names are presented here. (**c**) To test if the dataset samples were separated, we performed a principal component analysis (PCA) analysis on all protein expression data and as observed the CTL-Exo (in red) are clustered separately from the SHIV-Exo (in green). Unsupervised PCA plot is generated in R Studio by using median normalized and log_10_ transformed data from each sample type. CTL-Exo: Plasma exosomes isolated from uninfected RM; SHIV-Exo: Plasma exosomes isolated from SHIV-infected RM.

**Figure 3 viruses-15-00794-f003:**
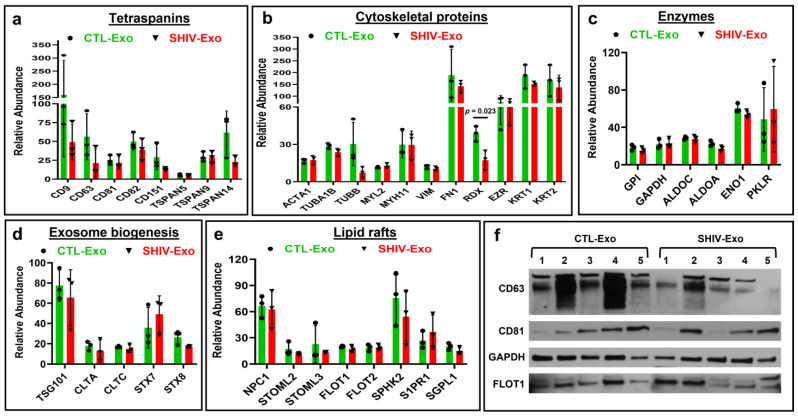
Hallmark exosomal proteins quantified by proteomic analysis in circulating exosomes of SHIV-infected and uninfected RM. (**a**–**e**) Relative abundance of quantified proteins in CTL-/SHIV-Exo. Proteins that exhibited group differences shown as bar graphs with green and red for CTL-Exo and SHIV-Exo, respectively. Plasma exosomes were isolated from three SHIV-infected and three uninfected RM (N = 3/group). Graphs show mean ± SD of relative abundance that was calculated from four experimental replicates of each sample. Protein group ‘b’ and ‘e’ passed the D’Agostino and Pearson normality test and protein group ‘c’ and ‘d’ passed the Shapiro–Wilk normality test. Protein group ‘a’ did not pass the normality test. The data sets were followed by unpaired t test with Welch correction, performed for normally distributed data. When the data did not pass the normality test, a non-parametric Mann–Whitney test was performed. Significant differences (*p* < 0.05) between groups are indicated. (**f**) The expression of CD63, CD81, GAPDH, and FLOT1 in CTL-/SHIV-Exo was further validated by Western blotting (N = 5/group). The protein coding gene names are presented here and were also described in the text. CTL-Exo: Plasma exosomes isolated from uninfected RM; SHIV-Exo: Plasma exosomes isolated from SHIV-infected RM.

**Figure 4 viruses-15-00794-f004:**
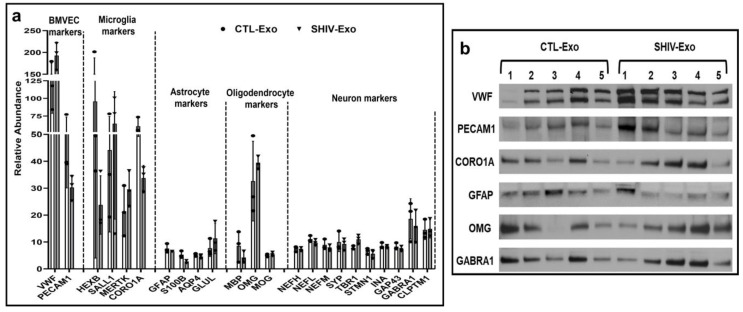
Relative abundance of CNS cell markers in circulating plasma exosomes. (**a**) Central nervous system (CNS) cells-specific exosomal proteins in CTL-/SHIV-Exo were quantified by proteomic analysis. Proteins that exhibited group differences shown as bar graphs with white and gray for CTL-Exo and SHIV-Exo, respectively. Plasma exosomes were isolated from three SHIV-infected and three uninfected RM (N = 3/group). Graphs show mean ± SD of relative abundance calculated from four experimental replicates per sample. For a protein group not passing the normality test, a non-parametric Mann–Whitney test was performed. There were no significant differences (*p* < 0.05) between groups. (**b**) The indicated proteins were further validated by Western blotting. For this assay, plasma exosomes were isolated from five SHIV-infected and five uninfected RM (N = 5/group). Protein coding gene names are presented here and were described in the text. CTL-Exo: Plasma exosomes isolated from uninfected RM; SHIV-Exo: Plasma exosomes isolated from SHIV-infected RM. BMVEC: Brain microvascular endothelial cells.

**Figure 5 viruses-15-00794-f005:**
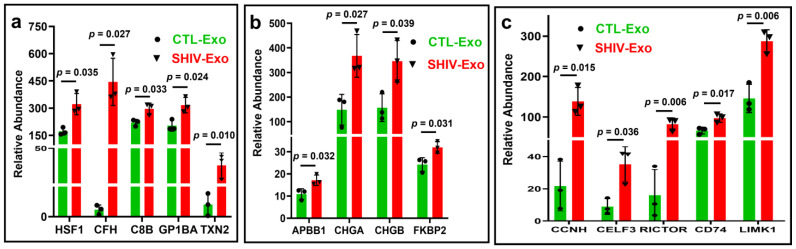
Increased expression of proteins involved in viral reactivation, inflammation, and neuropathology-associated interactive/signaling proteins in SHIV-Exo. (**a**–**c**) Relative protein abundance in CTL-/SHIV-Exo was quantified by proteomic analysis. Proteins that exhibited group differences are shown as bar graphs with green and red for CTL-Exo and SHIV-Exo, respectively. Plasma exosomes were isolated from three SHIV-infected and three uninfected RM (N = 3/group). Graphs show mean ± SD relative abundance that was calculated from four experimental replicates of each sample. All protein groups (**a**–**c**) passed the Shapiro–Wilk normality test, and the data sets followed by unpaired t test with Welch correction for normally distributed data. The protein coding gene names are presented here, and they were described in the text. CTL-Exo: Plasma exosomes isolated from uninfected RM; SHIV-Exo: Plasma exosomes isolated from SHIV-infected RM. q-values for: (**a**) all of them are 0.035; (**b**) all of them are 0.040; and (**c**) 0.022, 0.036, 0.017, 0.022, 0.017. Age-adjusted *p*-values were presented in the Appendix A.

**Figure 6 viruses-15-00794-f006:**
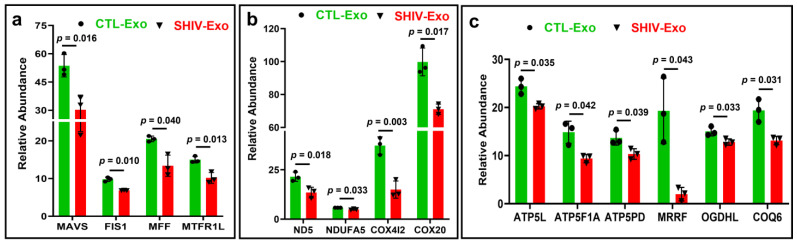
Decreased expression of proteins involved in mitochondrial biogenesis and ATP production in SHIV-Exo. (**a**–**c**) Differential expression of mitochondrial proteins in CTL-/SHIV-Exo quantified by proteomic analysis. Proteins that exhibited group differences are shown as bar graphs with green and red for CTL-Exo and SHIV-Exo, respectively. Plasma exosomes were isolated from three SHIV-infected and three uninfected RM (N = 3/group). Graphs show mean ± SD of relative abundance calculated from four experimental replicates/sample. Protein group ‘a’ and ‘c’ passed the Shapiro–Wilk normality test and protein group ‘b’ passed the D’Agostino and Pearson normality test. The data sets were followed by unpaired t test with Welch correction for normally distributed data. The protein coding gene names are presented here, and they were described in the text. CTL-Exo: Plasma exosomes isolated from uninfected RM; SHIV-Exo: Plasma exosomes isolated from SHIV-infected RM. q-values for: (**a**) 0.022, 0.022, 0.039, 0.022; (**b**) 0.024, 0.037, 0.013, 0.024; and (**c**) all of them are 0.043. Age-adjusted *p*-values were presented in the Appendix A.

**Figure 7 viruses-15-00794-f007:**
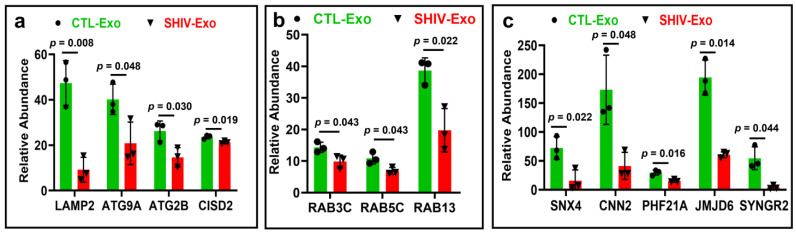
Decreased expression of proteins involved in autophagy, endosomal recycling, exocytosis, and sprouting angiogenesis in SHIV-Exo. (**a**–**c**) Differential expression of proteins involved in autophagy, endosomal recycling, exocytosis, and sprouting angiogenesis in CTL-/SHIV-Exo quantified by proteomic analysis. Proteins exhibiting group differences are shown as bar graphs with green and red for CTL-Exo and SHIV-Exo, respectively. Graphs show mean ± SD of relative abundance calculated from four experimental replicates/sample. All protein groups passed the Shapiro–Wilk normality test. The data sets were followed by unpaired t test with Welch correction for normally distributed data. The protein coding gene names are presented here and were described in the text. CTL-Exo: Plasma exosomes isolated from uninfected RM; SHIV-Exo: Plasma exosomes isolated from SHIV-infected RM (N = 3/group). q-values for: (**a**) 0.036, 0.048, 0.041, 0.038; (**b**) all of them are 0.042; and (**c**) 0.037, 0.048, 0.037, 0.037, 0.048. Age-adjusted *p*-values were presented in the Appendix A.

**Figure 8 viruses-15-00794-f008:**
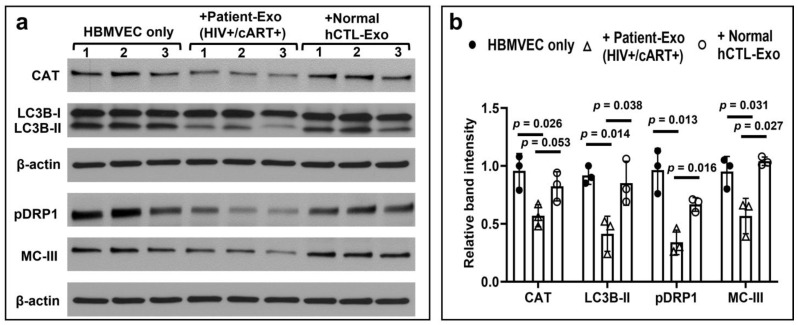
The effect of HIV-infected and cART-treated Patient-Exo on primary HBMVECs. (**a**) Cells were treated with 10 µg/mL of both Patient-Exo as well as exosomes isolated from healthy human plasma (hCTL-Exo) for 24 h. The indicated proteins were qualitatively detected by Western blotting from equal amount (15 µg) of clarified cell lysates and β-actin was used as an internal control. Three independent experiments were performed and are presented by Arabic numbers. (**b**) The relative band intensities of CAT, LC3B-II, pDRP1, and MC-III were compared by Image-J software (version 1.50). Values were mean ± standard deviation (SD). The data set passed the Shapiro–Wilk normality test and unpaired t test with Welch correction was performed for normally distributed data. q-values: all of them are 0.041 for HBMVEC only vs Patient-Exo; and 0.054, 0.043, 0.041, 0.041 for hCTL-Exo vs. Patient-Exo.

**Figure 9 viruses-15-00794-f009:**
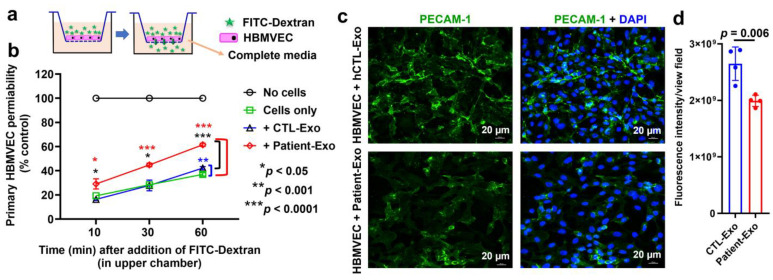
In vitro BBB permeability assays and the detection of PECAM-1 by immunofluorescence confocal microscopy in primary HBMVECs exposed to HIV+ Patient-Exo. (**a**) Schematic diagram of transwell migration-based BBB permeability in vitro assay. (**b**) Graph showing the BBB permeability efficiency of fluorescence labeled high molecular weight dextran (FITC-Dextran, 150 kDa) in the absence (black line) and presence (red, blue, and green lines) of cell monolayer. Cells without treatment (Cells only) or treated with normal human plasma exosomes (+CTL-Exo) were used as controls. The increased BBB permeability in cells treated with HIV-infected and cART-treated patient plasma exosomes (+Patient-Exo) is indicated by the red line. N = 4 experimental replicates were performed. (**c**) Representative immunofluorescent confocal microscope images of HBMVEC expressing PECAM-1 after 24 h treatment (10 µg/mL) with either hCTL-Exo or Patient-Exo. (**d**) The relative fluorescence intensity of PECAM-1 per view field was compared.

**Figure 10 viruses-15-00794-f010:**
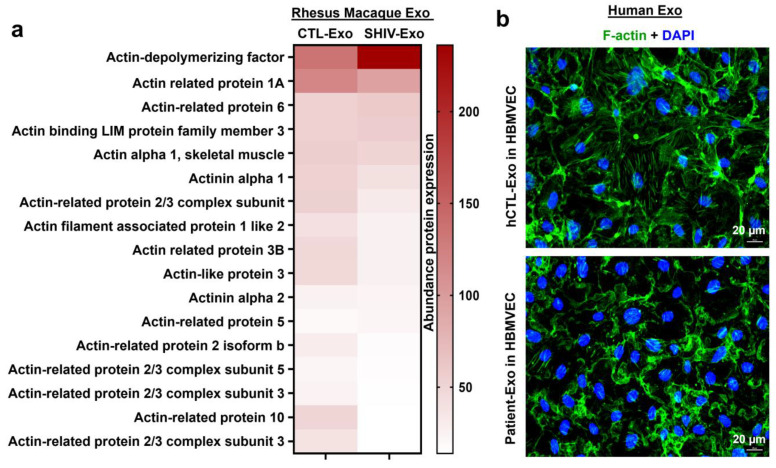
Differential expression of actin cytoskeleton proteins in RM plasma exosomes and the expression pattern of F-actin in primary HBMVECs after exposure to HIV+ Patient-Exo. (**a**) The heat map of the abundance actin cytoskeleton proteins quantified by proteomic analysis was generated by using GraphPad Prism version 9.0.0 for Windows. (**b**) Representative immunofluorescent confocal microscope images of the HBMVEC expressing F-actin after 24 h treatment (10 µg/mL) with either hCTL-Exo or Patient-Exo.

## Data Availability

The datasets generated during and/or analyzed during the current study are available from the corresponding author on reasonable request.

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
