# Peer review of "Circulating Plasma Exosomal Proteins of Either SHIV-Infected Rhesus Macaque or HIV-Infected Patient Indicates a Link to Neuropathogenesis"

_viruses, 2023, doi:10.3390/v15030794_

Round 1

Reviewer 1 Report

This manuscripts presents proteomics data in the context of exosomal biomarkers for HIV-associated neurocognitive disorder (HAND), which has been a common comorbidity for persons living with HIV infection. The study design has a main strength in the inclusion of SHIV-infected and rhesus macaques and uninfected control animals, as well as human samples in multifaceted analyses. Several weaknesses in data analyses, as summarized below, compromise the conclusions drawn from this study.

Main issues

1. The use of p <0.05 as a threshold for statistical significance is too liberal. Given the large number of proteins analyzed (compared between groups, each p value must be accompanied by a q value. If an FDR of <1% is deemed robust (as stated in the methods section), the corresponding p value must be well below 0.05.

2. Most of the individual proteins being highlighted in the figures (e.g., Figure 3 for simian samples and Figure 8 for human samples) show modest differences between comparison groups. Such minor differences could easily be obscured by fluctuations over time, so there should be at least some longitudinal sampling to ensure that the cross-sectional results being observed here can be replicated at different time points.

3. SHIV-infected and uninfected macaques differ in age (11.8 years versus 8.7 years), and therefore the comparisons are not that meaningful without statistical adjustment for age.

4. For comparison of human samples, demographic factors should be treated as covariates in all analysis, as age, sex and duration of HIV infection and duration of antiretroviral therapy can all alter the plasma (EV) proteomics profiles.

5. The top 50 hits in either directions (up- or down-regulation) are worth noting. Perhaps a principal component analysis can offer an extra dimension for the presentation of key findings.

6. The proteins being highlighted in this study do not include CCL2/MCP-1, a well-known marker for HAND. This point deserves some discussion.  

Author Response

Response to the Reviewer

Dear Reviewer 1,

We truly thank you for your careful review of our manuscript and constructive comments and have used this feedback to significantly revise (and, we believe, substantially improve) the manuscript.  We appreciate your general comment about the strength of our study design. Please find below our point-by-point italicized responses (highlighted in the revised manuscript) to your critiques and comments.  According to the reviewer’s recommendation, we have added a biostatistician (Jeffery G Shaffer) as co-author in our revised manuscript to address several statistical comments below. We do not believe that any of the changes in the revised manuscript altered the general conclusion or results presented in the original manuscript.

Specific comments:

  1. The use of p <0.05 as a threshold for statistical significance is too liberal. Given the large number of proteins analyzed (compared between groups, each p value must be accompanied by a q value. If an FDR of <1% is deemed robust (as stated in the methods section), the corresponding p value must be well below 0.05.

Response: We thank the Reviewer for this observation and have included the q-values as suggested. We showed the actual p-values in the revised figures and listed the q-values in the figure legends for figures 5–8. We also tabulated the q-values and included them as supplementary information. Whenever possible, we included the q-values described by Storey, 2002 (J. R. Statist. Soc. B, 2002, 64, Part 3, pp. 479–498). For those figures or panels with too few comparisons for determining the proportion of true null hypotheses as required for Storey q-values, we used the adjusted p-values according to Benjamini and Hochberg (1995).

  1. Most of the individual proteins being highlighted in the figures (e.g., Figure 3 for simian samples and Figure 8 for human samples) show modest differences between comparison groups. Such minor differences could easily be obscured by fluctuations over time, so there should be at least some longitudinal sampling to ensure that the cross-sectional results being observed here can be replicated at different time points.

Response: In Figure 3, we wanted to show that several hallmark exosomal proteins were successfully quantified by our proteomic analysis of crExo in both SHIV-infected and uninfected RM. This observation was further supported by our western blot analysis. Similarly, in Figure 4, we attempted to show that a number of CNS cell-specific markers were able to detect both in SHIV-/CTL-Exo. These data also further supported by western blot analysis. In Figure 3 and in Figure 4, we had no intention to show the significant differences between comparison groups. From Figure 5 to Figure 7, we compared the significant (p < 0.05) differentially expressed proteins between SHIV-Exo and CTL-Exo that are involved in important biological functions and signaling pathways.

We agree with the reviewer that in Figure 5-7 small significant differences for some proteins could be due to veriations over the period of time. In the method sections (sub-section: 2.1 SHIV-infected and cART-treated rhesus macaque plasma) we mentioned that plasma samples were collected at the time of necropsy. Therefore, we do not have longitudinal samples for these animals. This limitation was mentioned in the “Limitations and potentials of their research” section in our revised manuscript.    

  1. SHIV-infected and uninfected macaques differ in age (11.8 years versus 8.7 years), and therefore the comparisons are not that meaningful without statistical adjustment for age.

Response: We appreciate this observation and agree with the Reviewer that age-adjusted p-values could potentially be informative. To this end, we calculated the age-adjusted p-values using an analysis of covariance approach.  The age-adjusted p values are now provided with the unadjusted p values as Supplementary Data 1 and are also indicated in result sections of Figures 5–7.  

  1. For comparison of human samples, demographic factors should be treated as covariates in all analysis, as age, sex and duration of HIV infection and duration of antiretroviral therapy can all alter the plasma (EV) proteomics profiles.

Response: As a proof-of -concept study, we included only one commercially available patient plasma in this study to compare the effect of HIV-infected and cART-treated Patient-Exo with one healthy control Exo (detailed was mentioned in Method section) in primary HBMVECs. We performed three independent experiments indicated by Arabic numbers mentioned in Figure 8. Therefore, in this case, we don’t have option for covariant analysis with age, sex, duration of HIV-infection, and duration of cART therapy. This limitation was mentiond in the “Limitations and potentials of their research” section in our revised manuscript.

  1. The top 50 hits in either directions (up- or down-regulation) are worth noting. Perhaps a principal component analysis can offer an extra dimension for the presentation of key findings.

Response: We performed principal component analysis (PCA) on all protein expression data and as observed the CTL-Exo are clustered separately from the SHIV-Exo. Unsupervised PCA plot is generated in R Studio by using median normalized and log10 transformed data from each sample type and this new figure is included as Figure 2c in our revised manuscript.

  1. The proteins being highlighted in this study do not include CCL2/MCP-1, a well-known marker for HAND. This point deserves some discussion.

Response: In our proteomic analysis, monocyte chemoattractant protein-1 (CCL2/MCP-1) was not quantified either control or SHIV-infected plasma exosomes. This point was discussed with references (Ref # 109-111) in the “Limitations and potentials of their research” section in our revised manuscript.

Reviewer 2 Report

While this is baasically agoo experiemental and clinical paper, there are some weaknesses that need to be addressed and there is still room for improvement. The authors have justifiably tried to show the link between cellular physiology and HIV. While doing so, they presume that all HIV clades and variants are likely to behave the same way. They also ignored the virion physiology of HIV and the various roles of the viral proteins pertaining to the ability of the virus in entering the brain.

1) Much is known about the roles of viral proteins in entering important organs such as the brain and placenta. An examples is:

https://pubmed.ncbi.nlm.nih.gov/31072073/

https://pubmed.ncbi.nlm.nih.gov/31698857/

The two papers are saying that in many viruses including HIV and flaiviviruses such as Zika, disorder at the outer shell (matrix) is responsible for viral evasion agains antibodies and the ability of viral penetration into organs, whereas the disorder at the inner shell ( nucleocpascid (NC)) is responsible for the rapid replication of the virus in its attemp to evade the host immune system. While the NC could have some effects on the valr entry into the brain by producing a large number of particles, the matrix has a more direct effect since disorder allows more efficient binding to proteins especially host proteins.  Goh et al offered clues rom various viruses. Many HIV variants have the highest matrix disorder among nearly all viruses. An exception is HSV. Both vaccines for both HIV and HSV have never been discovered and both are highly adapted in hiding in the body specifically in the brain. A second piece of evidence offered can be found in the second paper by Goh et al. In this paper it shows that the highly pathogenic Yellow Fever that kills 50% of infected person by easily entering vital organs has higher disorder at both inner and outer shells. But another ,uch milder cousing that is even milder than any dengue virus but has a unique ability to enter the placenta has a smimilarly high disorder at the outer shell but much lower disorder at its inner shell.. The two paper provides sufficient evidence that disorder at the outer shell plays an important role in viral penetration into organs especially the brain. Such  has not been mentioned in the paper being reviewed.

2) What is unknown is: How does the viral proteins, namely matrix interacts with the host proteins? Do the data presented by the authors offer any suggestions for such mechanism?

3) We  need to be careful about extrapolating the behaviors and properties of SHIV ot HIV in general. SHV refers to a specific variant that was engineered to infect primates such as macque but HIV in general refers to many HIV variants. As mentioned in (1), a large number of HIV variants, not all, have been found to have extemrly high matrix disorder. We cannot therefore assume that SHV will have the same properties as every other HIV variants

4) With (1)-(3) in mind, the authors should list out the limitations and potentials of their research  while mentioning what  esle need to be done .

.

Author Response

Response to the Reviewer

Dear Reviewer #2

We appreciate your general comments that we have justifiably tried to show the link between cellular physiology and HIV. Please find below our point-by-point italicized responses (highlighted in the revised manuscript) to your critiques and comments.   

Specific comments:

1) Much is known about the roles of viral proteins in entering important organs such as the brain and placenta. An examples is:

https://pubmed.ncbi.nlm.nih.gov/31072073/

https://pubmed.ncbi.nlm.nih.gov/31698857/

The two papers are saying that in many viruses including HIV and flaiviviruses such as Zika, disorder at the outer shell (matrix) is responsible for viral evasion agains antibodies and the ability of viral penetration into organs, whereas the disorder at the inner shell ( nucleocpascid (NC)) is responsible for the rapid replication of the virus in its attemp to evade the host immune system. While the NC could have some effects on the valr entry into the brain by producing a large number of particles, the matrix has a more direct effect since disorder allows more efficient binding to proteins especially host proteins.  Goh et al offered clues rom various viruses. Many HIV variants have the highest matrix disorder among nearly all viruses. An exception is HSV. Both vaccines for both HIV and HSV have never been discovered and both are highly adapted in hiding in the body specifically in the brain. A second piece of evidence offered can be found in the second paper by Goh et al. In this paper it shows that the highly pathogenic Yellow Fever that kills 50% of infected person by easily entering vital organs has higher disorder at both inner and outer shells. But another ,uch milder cousing that is even milder than any dengue virus but has a unique ability to enter the placenta has a smimilarly high disorder at the outer shell but much lower disorder at its inner shell.. The two paper provides sufficient evidence that disorder at the outer shell plays an important role in viral penetration into organs especially the brain. Such has not been mentioned in the paper being reviewed.

Response: Thank you for emphasizing this point by providing the important references. In the “Limitations and potentials of their research” section, we mentioned that several viral proteins especially, HIV-1 Matrix protein (Ref # 106-107) plays an important role in viral evasion against antibodies and the ability of viral penetration into the brain, which induced HIV-associated neuropathogenesis. Recently, Caccuri et al. (2022) showed the ability of the HIV-1 Matrix protein p17 to cross the BBB and to reach the CNS, thus possibly playing a crucial role in neuronal dysfunction in HAND (Ref # 108).

2) What is unknown is: How does the viral proteins, namely matrix interacts with the host proteins? Do the data presented by the authors offer any suggestions for such mechanism?

Response: We agree with the Reviewer that studies on how viral proteins interacts with several host proteins and accelerate the development and progression of HIV-associated neuropathogenesis is limited. Interestingly, in our data set, ‘All protein Reactome’ analysis indicats that several host proteins involved in Vif-mediated degradation of APOBEC3G, Vpu mediated degradation of CD4, Rev-mediated nuclear export of HIV RNA, Tat-mediated elongation of the HIV-1 transcript, Vpr-mediated nuclear import of Pre-Integration Complex, and formation of the HIV-1 early elongation complex (unpublish data). We have decided to publish this interesting crExo-containing host-virus interactive proteins and their mechanism(s) of action on the development and progression of HIV-associated neuropathogenesis as a continuation of our present manuscript where we will mainly focus on the host-virus interactive proteins, which are possibly linked with neuropathogenesis.

3) We  need to be careful about extrapolating the behaviors and properties of SHIV ot HIV in general. SHV refers to a specific variant that was engineered to infect primates such as macque but HIV in general refers to many HIV variants. As mentioned in (1), a large number of HIV variants, not all, have been found to have extemrly high matrix disorder. We cannot therefore assume that SHV will have the same properties as every other HIV variants.

Response: We appreciate the Reviwer for mentioning the possible differences in the behaviors and properties of SHIV and HIV varitents. We have included this information in the “Limitations and potentials of their research” section.

4) With (1)-(3) in mind, the authors should list out the limitations and potentials of their research  while mentioning what  esle need to be done.

Response: We included all of the above mentiond points in the “Limitations and potentials of their research” section in our revised manuscript.

Round 2

Reviewer 1 Report

The revisions and responses to my earlier critiques are adequate enough that I have no further concerns.

Reviewer 2 Report

Improvement seen.